# Assessment of the Tumor–Stroma Ratio and Tumor-Infiltrating Lymphocytes in Colorectal Cancer: Inter-Observer Agreement Evaluation

**DOI:** 10.3390/diagnostics13142339

**Published:** 2023-07-11

**Authors:** Azar Kazemi, Masoumeh Gharib, Nema Mohamadian Roshan, Shirin Taraz Jamshidi, Fabian Stögbauer, Saeid Eslami, Peter J. Schüffler

**Affiliations:** 1Institute of General and Surgical Pathology, Technical University of Munich, 81675 Munich, Germany; azar.kazemi@tum.de (A.K.); fabian.stoegbauer@tum.de (F.S.); peter.schueffler@tum.de (P.J.S.); 2Department of Medical Informatics, School of Medicine, Mashhad University of Medical Sciences, Mashhad 9177948564, Iran; 3Department of Pathology, Faculty of Medicine, Mashhad University of Medical Sciences, Mashhad 9137913316, Iranroshann@mums.ac.ir (N.M.R.); tjamshidish@mums.ac.ir (S.T.J.); 4Pharmaceutical Sciences Research Center, Institute of Pharmaceutical Technology, Mashhad University of Medical Sciences, Mashhad 9177948954, Iran; 5Department of Medical Informatics, University of Amsterdam, 1105 AZ Amsterdam, The Netherlands

**Keywords:** tumor-infiltrating lymphocytes, tumor–stroma ratio, colorectal cancer, inter-observer

## Abstract

Background: To implement the new marker in clinical practice, reliability assessment, validation, and standardization of utilization must be applied. This study evaluated the reliability of tumor-infiltrating lymphocytes (TILs) and tumor-stroma ratio (TSR) assessment through conventional microscopy by comparing observers’ estimations. Methods: Intratumoral and tumor-front stromal TILs, and TSR, were assessed by three pathologists using 86 CRC HE slides. TSR and TILs were categorized using one and four different proposed cutoff systems, respectively, and agreement was assessed using the intraclass coefficient (ICC) and Cohen’s kappa statistics. Pairwise evaluation of agreement was performed using the Fleiss kappa statistic and the concordance rate and it was visualized by Bland–Altman plots. To investigate the association between biomarkers and patient data, Pearson’s correlation analysis was applied. Results: For the evaluation of intratumoral stromal TILs, ICC of 0.505 (95% CI: 0.35–0.64) was obtained, kappa values were in the range of 0.21 to 0.38, and concordance rates in the range of 0.61 to 0.72. For the evaluation of tumor-front TILs, ICC was 0.52 (95% CI: 0.32–0.67), the overall kappa value ranged from 0.24 to 0.30, and the concordance rate ranged from 0.66 to 0.72. For estimating the TSR, the ICC was 0.48 (95% CI: 0.35–0.60), the kappa value was 0.49 and the concordance rate was 0.76. We observed a significant correlation between tumor grade and the median of TSR (0.29 (95% CI: 0.032–0.51), *p*-value = 0.03). Conclusions: The agreement between pathologists in estimating these markers corresponds to poor-to-moderate agreement; implementing immune scores in daily practice requires more concentration in inter-observer agreements.

## 1. Introduction

Diagnosis and treatment decisions are vital, and prognosis prediction is crucial for enhancing clinical outcomes in patients with CRC [1]. Although the incidence of CRC is stable or decreasing in some countries, Asian and European countries show an increasing trend in CRC incidence [2]. Also, incidence and mortality arising from CRC are increasing among young people around the world [2]. The survival of CRC patients is improved in countries where screening programs are implemented but differs in countries with low-standing screening programs [3].

Despite these improvements, CRC remains one of the deadliest cancers for patients [4]. Therefore, questions about early diagnosis and more accurate prognosis have forced the study of new prognostic biomarkers. Tumor microenvironment assessment and immune cell detection have grown following clinical success trends in therapeutic strategies for some solid malignancies. TILs and the TSR are two new biomarkers that convey crucial information for the diagnosis, prognosis, and treatment response prediction of various solid cancers [5,6,7,8,9].

TILs that show a host response to the tumor are stronger predictors of patient survival than TNM classification [10]. Therefore, they can be used as a prognostic factor across solid tumor types, including CRC [11,12,13]. The International TILs Working Group (ITILsWG) standardized the evaluation of TILs on HE-stained sections in breast cancer [14] and recommended using the same methodology for other solid malignancies to incorporate TIL quantification into clinical practice and research.

Currently, there is no validated cutoff system to report TILs in clinical practice. Therefore, recent studies should contribute to determining the safe threshold to provide more evidence for introducing an ideal cutoff system for daily practice [15]. However, according to ITILsWG recommendations [16], this new biomarker has been reported as a continuous variable. To implement a clinically relevant cutoff system, researchers who have applied the ITILsWG standardized scoring methodology have investigated different cutoffs to classify TILs into low- to high-grade categories in colorectal [17,18] and breast cancers [14,19] and have evaluated the prognostic value of TILs based on these cutoff systems. Iseki et al. [17] evaluated the ITILsWG method to standardize the TIL estimation and prognostic utility assessment of TILs in CRC. Their study reported that the methodology proposed by the ITILsWG can be useful in assessing TILs in CRC and can be used as a prognostic factor. A recent study by Fuchs et al. [18] validated the efficacy of the ITILsWG methodology in a large cohort of CRC patients. The study showed that compared with intraepithelial TILs, stromal TILs assessed using the ITILsWG approach have a better prognostic value in CRC. As these cells are dense in some parts of the tissue, visual assessment by pathologists is prone to error [20]. The study assessed inter-observer agreement in face-to-face and non-intensive training models and reported good and moderate inter-observer agreements, respectively [18].

TSR, as a percentage of the stroma area compared to the tumor cell area, is a histological feature assessed based on the morphological evaluation of HE sections. TSR, which indicates the quantity of the stromal component surrounding cancer cells in a tissue section, has been identified as a determinant of progression [21] and a predictor of prognosis in CRC [22,23]. Based on a meta-analysis [22], in most studies that validated the prognostic value of TSR in patients with CRC, a higher percentage of stroma (>50%) was associated with a worse prognosis. 

These studies suggest that estimating stromal components and investigating new biomarkers can be helpful in clinical practice by incorporating these bioscores into diagnosis and treatment strategies. However, concerns have been expressed regarding the reliability of the approximation of stromal component entities. Although few studies have shown the efficacy of proposed ITILsWG guidelines for predicting outcomes in CRC, the concordance rates in quantifying TILs showed variable agreement [18]. Likewise, semi-quantitative reporting of TSR is under study to investigate the variation in concordance rates. Recently, Souza da Silva et al. [23] showed that TSR estimation with good concordance was feasible. Our study aimed to determine the inter-observer agreement of TSR assessment and TIL scoring in CRC HE-stained sections using the proposed methods.

## 2. Material and Methods

### 2.1. Patients

This retrospective study was conducted on patients who underwent surgical resection for grades II and III of colorectal adenocarcinoma between 2010 and 2017 in the Omid oncology hospital affiliated with the Mashhad University of Medical Sciences, Mashhad, Iran. All patients in the study signed an informed consent for using their data and tissues in the research. All the procedures in the study were performed according to the institutional and national ethical standards and rules of the Declaration of Helsinki of 1975 and later comments. This study was approved by the Mashhad University Medical Sciences Research Ethics Committee (approval ID: IR.MUMS.MEDICAL.REC.1399.557).

Standardizing the detection and quantification of biomarkers on HE slides, as a cost-effective and widely available staining method, can greatly benefit diagnosis procedures, particularly in developing countries. We utilized the confirmed CRC data gathered by a previously conducted study [24]. One hundred and one cases with histological features of colorectal adenocarcinoma and available HE-stained sections were collected from the pathology department archives. HE slides were reviewed via light microscopy by a pathologist to select one representative HE slide (4 µm thick) for each patient. Patients were excluded if they were treated before surgery, if no tumor mass or invasion was found in the tissue section, or if the tumor was entirely necrotic. As a result, the final study cohort comprised 86 cases, all of which had assessable HE slides. Three senior pathologists (M.G, N.M.R, S.T.J) with a range of 8 to 17 years of clinical and surgical pathology experience participated in the study. Two pathologists (observer 1: OBS1 and observer 2: OBS2) were experienced in TIL scoring and the other pathologist (observer 3: OBS3) had less experience in TIL counting. No grouping training session was provided, but the ITILsWG guidelines and materials were provided to participating pathologists [15] and any questions regarding observation and scoring TILs and TSR were clarified by OBS1. Figure 1 shows the study method.

According to the ITILsWG’s proposed methodology of 2014 [14], the density of TILs on HE slides was evaluated. As the study recommended, visual measuring of stromal TILs, which are a superior parameter, using standard HE-stained sections is considered as a primary aim of TIL quantification. In summary, mononuclear cells, including lymphocytes and plasma cells within the stromal area, should be evaluated. Other cells, such as granulocytes and polymorphonuclear leukocytes, should be excluded. TILs outside of invasive tumor boundaries should also be excluded. The average stromal TIL density within the borders of the invasive tumor over the entire whole HE section should be scored. Crush artifacts, necrosis, or fibrosis should be excluded. Figure 1A–C show three levels of TIL density according to the methodology.

In a blinded manner, three observers independently estimated the density of TILs in the invasive tumor front and intratumoral TILs. To avoid any bias, all pathologists were blinded to other clinical and histopathological details and then stored the scores in a columnar form created for reporting. The scores were reported as continuous values and rounded to the nearest 0.05. All reports were collected by a data analyzer who was blinded to other patients’ data during the observation phase. All results were kept confidential, with no feedback to the participants. Using the ITILsWG method [14] and the cutoffs suggested in previous studies [17,18,19], the continuous TIL scores were categorized into two or three grades: low to high. The suggested grading systems for the quantification of intratumoral and tumor-front stromal TILs are shown in Table 1 and Table 2.

### 2.2. Tumor–Stroma Ratio Estimation

TSR was quantified visually by three pathologists on the tissue slides used in routine diagnostic pathology using conventional light microscopy according to the approximation of TSR in the Sullivan et al. study [25]. The estimations were recorded as a tumor-to-stroma ratio in a blinded manner, and inter-observer agreement in the estimation of the TSR was assessed by three pathologists. In areas of necrosis, artifacts, and crush artifacts, stromal cells were not estimated. According to previous studies, which were reviewed in a meta-analysis by Zhu et al. [22], a cutoff of 50% was used for TSR scores. Cases with ≤50% stroma were considered stroma-poor, and cases with >50% stroma were considered stroma-rich. Two examples of stroma-poor and stroma-rich cases are shown in Figure 1D,E.

### 2.3. Statistical Analysis

Statistical analyses were performed using R version 4.2.1. Missing values were imputed using the mean of sample scores. The intra-class correlation coefficient (ICC) was calculated using a mixed model to assess the inter-observer agreement for the continuous scores of TILs and TSR. After categorizing the scores, Cohen’s kappa was used for pairwise inter-observer reliability between two observers, and Fleiss’ kappa as an adaptation of Cohen’s kappa [26] was used to measure inter-rater agreement among all pathologists.

Concordance rate analysis was performed as the overall agreement between each observer and the median value for each sample and the Bland–Altman plots were drowned to compare the intratumoral TIL scores’ discordance. To assess the pairwise correlation between pathologists, the Spearman correlation coefficient was applied to study the raw biomarker scores because the scores were not normally distributed. The McNemar test was performed to determine whether there was a difference in the frequency of categories scored by the two pathologists. Furthermore, we assumed that the sample was difficult to estimate if all pathologists disagreed (giving extremely different scores to a sample) and showed the differences through a heatmap graph. Pearson’s correlation analysis was used to study the association between biomarkers calculated by pathologists and patient data. Statistical significance was set at *p*-value ≤ 0.05.

## 3. Results

The study population included 86 patients aged between 21 and 87 years, with a mean age of 56.74 years. Of the patients, 54.65% were male.

### 3.1. Intratumoral Stromal TILs

The ICC for estimating intratumoral TILs was 0.505 (95% CI: 0.35–0.64) (Table 1). The overall kappa values for intratumoral stromal TIL evaluation using different cutoff systems were in the range of 0.21 to 0.38 (Table 1). The concordance rates show the overall agreement among each pathologist’s estimation for each case compared with the same case’s intratumoral TIL median value. The overall concordance for quantifying the density of TILs in the intratumoral ranged from 0.61 to 0.72 (Table 1). 

To visualize the discordance among pathologists based on the TIL density scores estimated by each observer and OBS_Median (median calculated for each sample as a gold standard observer), Bland–Altman plots were drawn (Figure 2A–C). While OBS2 and OBS3 tended to estimate intratumoral TILs densities lower than OBS_median, OBS1 gave higher TIL scores than OBS_Median (Figure 2A–C and Figure 3A).

The consensus for different cutoff categories of TILs is shown in Appendix A. A positive consensus meant that all pathologists agreed on the TIL score. Likewise, a partially positive consensus meant that two of the three pathologists agreed, and a negative consensus can be interpreted as a complete disagreement between all pathologists in evaluating the same case. The frequencies of cases in each category of intratumoral stromal TILs recorded by the pathologists are shown in Appendix A. The results indicate that compared to three-tiered systems, using two-tiered systems caused more positive consensus among pathologists.

Correlations between pathologists were in a range of 0.48 to 0.64 (Table 2). Pairwise agreement evaluation for estimation of stromal TILs using different cutoff systems showed that there was a significant difference between OBS1 and the other two pathologists. The pairwise consensus showed the percentage of cases in which the two pathologists agreed (Appendix A). A heatmap graph was drawn to represent slides that were difficult to score and caused under- or overestimation of TILs by pathologists. Figure 4A shows that for the same cases, some observers reported scores higher than those of other observers.

### 3.2. TILs in the Tumor Front

The intraclass coefficient, kappa statistic, and concordance rate among observers for estimating the tumor-front TILs were calculated according to different cutoff systems, and the results are shown in Table 3.

The ICC for evaluation of the stromal TILs in the tumor front was 0.52 (95% CI: 0.32–0.67). The overall kappa value for the different categories of TIL density in the tumor front ranged from 0.24 to 0.30. The overall agreement between each pathologist’s estimation and the TIL median value for the same case in different categories ranged from 0.66 to 0.72. In determining the number of slides that were more difficult to evaluate regarding the stromal TILs in the tumor front, the consensus for different cutoff categories of stromal TILs in the tumor front shows that pathologists had consensus on at least 37% of cases. According to the recorded TILs in the tumor front, at least 58% of cases were estimated by OBS1 categorized in high density of TILs (Appendix A).

The correlation among pathologists showed that the correlation between TILs scored by OBS2 and OBS3 was higher than the correlation with TILs scored by OBS1 (Table 2). There were no significant differences between OBS2 and OBS3 in all grading systems for scoring TILs in the tumor front (Appendix A). Kappa values ranging from 0.16 to 0.47 were obtained. According to System 4, which utilizes a two-tiered categorization, and in comparison with other cut-off systems, a higher agreement was observed between OBS1 and OBS3 who are experienced in scoring TILs (consensus: 0.66 (95% CI: 0.55–0.76); kappa value: 0.40) (Appendix A). The heatmap indicates the challenging slides for evaluation that led pathologists to under- or overestimate stromal TILs in the tumor front (Figure 4B). It seems OBS1 estimated the tumor-front TILs with a higher threshold, and patterns of tumor-front TILs estimated by OBS2 and OBS3 showed that they gave lower scores (Figure 2D–F and Figure 3B). In total, OBS1 estimated the TILs with a higher threshold. 

### 3.3. Tumor–Stroma Ratio

According to the Spearman coefficient analysis results, for raw continuous TSR scores (Table 2), the TSR scored by OBS1 correlated with the TSR scores estimated by OBS2. Stromal estimations were divided into two categories: stroma-rich and stroma-poor. Evaluation of TSR led to an ICC of 0.48 (95% CI: 0.35–0.60), and an overall kappa value of 0.49. However, the concordance rate between pathologists was 0.76, which was good (Table 4). Furthermore, a positive consensus existed over the 75 slides for estimating TSR (Appendix A). In addition, the table shows the frequency of cases estimated by the pathologists in each category.

The agreement between pathologists for estimating the TSR ranged from 0.26 to 0.72 and there was no significant difference between OBS1 and OBS2 (consensus: 0.95, kappa value: 0.72) (Appendix A). The heatmap for TSR estimation illustrates how some observers assessed TSR differently from other observers. It was assumed that samples with disparate scores could not be easily estimated (Figure 3C). The distribution of the TSR scores by pathologists for the 86 slides is shown in Figure 3C. The distribution of TSR scored by OBS1 and OBS3 is as close to OBS_Median, but OBS2 estimated the TSR to be relatively lower than OBS_Median. Additionally, Figure 2G–I, which visualize discordance among pathologists for estimating TSR, show that the variability in stroma-rich cases was higher than that in stroma-poor cases.

### 3.4. Correlation between Scored Biomarkers and Patient Data

The correlations between median biomarker values and patient data are presented in Table 5. The median values of intratumoral and tumor-front TIL scores and tumor–stroma ratio scored by pathologists were calculated to investigate the correlation with patient age, sex, and tumor grade. There was a poor correlation between tumor grade and TSR (0.29 (95% CI: 0.032–0.51), *p*-value = 0.03). There was no evidence of a correlation between the TIL scores and patient data.

## 4. Discussion

The current study evaluated inter-observer agreement among pathologists estimating the density of stromal TILs in intratumoral and tumor-front compartments and the tumor–stroma ratio in CRC using the ITILsWG recommended methodology and proposed cutoff values in previous studies. In the current study, HE-stained slides, as the most common histological staining for cancer diagnostic purposes around the world, were reviewed to measure stromal TILs and TSR. 

Within our team, two pathologists (OBS1 and OBS3) were specifically experienced in TIL scoring, while OBS2 had less experience in diagnostics and counting TILs. We observed a correlation between the scores recorded by OBS2 and the scores recorded by OBS3, who had greater diagnostic experience and previous exposure to TIL quantification, and the results showed for estimation of TSR, the ratio of tumor–stroma were estimated by OBS1 and OBS2 were more correlated.

### 4.1. Intratumoral Stromal TILs

According to the criteria of Koo and Li [27], the ICC for assessing intratumoral stromal TILs can be interpreted as poor-to-moderate agreement. According to these criteria, the 95% confidence interval should be considered as a basis to evaluate the reliability level which categorizes values <0.5 as poor, 0.5 to 0.75 as moderate, 0.75 to 0.9 as good, and values greater than 0.9 as excellent reliability [27]. According to Landis and Koch’s interpretation criteria [28], the overall kappa values were interpreted as a fair agreement. According to these criteria, the value of zero is interpreted as poor agreement, 0 to 0.2 as slight, 0.2 to 0.4 fair, 0.4 to 0.6 as moderate, 0.6 to 0.8 substantial, and 0.8 to 1 as perfect agreement. The results indicate a lower agreement was obtained if we categorized the intratumoral TIL densities into three-tiered categories and would improve to a higher inter-observer agreement (kappa value: 0.38, concordance rate: 0.72, and consensus rate: 55.81%) when categorizing TILs into two-tiered groups. Tramm et al. [19] also showed that when categorizing TIL density into two levels, the inter-observer agreement would increase.

The pairwise evaluation indicated fair to moderate agreement among pathologists estimating intratumoral TILs using the ITILsWG proposed methodology cutoffs (System 1) [14] and Iseki et al. [17]’s suggested cutoff (System 4). There was a moderate agreement between OBS2 and OBS3 in 72.1% of cases (kappa value: 0.44) using the ITILsWG cutoff system (System 1) and they agreed on TIL scores of 74.4% of cases (kappa value: 0.43) categorized according to the suggested cutoff System 4. There was no evidence of significant differences between OBS2 and OBS3 in estimating intratumoral TIL scores. The Spearman correlation coefficient for scoring intratumoral TILs was moderately positive among the pathologists. However, the TIL values scored by OBS2 and OBS3 were more correlated. All evidence indicates that scores recorded by OBS1, who had experience in training with the ITILsWG methodology for estimating the density of TILs, were significantly different from those of the two other pathologists. Overall, the variability in cases with an intermediate density was higher than that in cases with a low or high density of TILs.

This study shows that in cases with a moderate mean density of intratumoral TILs, pathologists deal with challenges in scoring these cases, and TIL densities are variably estimated. However, tissues with low or high mean densities of TILs were estimated in the same range. The current study emphasizes the results of a previous study [19] showing that the source of inter-observer disagreement could be the variable threshold between pathologists. Obtaining aid from software in the agreement study process is a probable solution for dealing with disagreement factors. Denkert et al. [29] have shown that although using software in the assessment of TILs makes the assessment approach complex, optimizing the reproducibility is likely.

In some studies [5,14,15,17,19,23,30], the suggested cutoffs for grading stromal TILs have varied, and in most of these studies, the cutoff was estimated for grading TILs in breast cancer [5,15,19,30]. Comparing different cutoff systems categorizing the TILs, although none of the cutoff systems showed a moderate agreement among all pathologists, the ITILsWG [14] suggested cutoff values (<10%, 10–50%, >50%) and Iseki et al. [17] suggested cutoff values (<42%, ≥42%) were more reliable. The concordance for each pathologist also improved when we used the two-tiered system (system 4), which was suggested based on a CRC study [17], indicating that categorizing tumors with two levels of TILs as low vs. high can facilitate the assessment of TILs in practice. However, according to Denkert et al. [29], the first ring of the inter-observer agreement study, in which 32 pathologists participated in scoring the TILs in 60 breast cancer cases, showed agreement between pathologists in comparison with dichotomous grading systems. According to their study, grading TILs into two-tiered cutoffs does not guarantee better agreement, and setting precise cutoffs is crucial. In our study, dichotomous cutoff groups were used according to the CRC study [17], which slightly improved the overall and pairwise inter-observer agreement.

Furthermore, this study is in accordance with previous studies [17,18] that showed that although the methodology recommended by the ITILsWG is proposed for scoring TILs in breast cancer, using the methodology for detecting and assessing TILs can be useful in CRC. However, according to our results, the suggested cutoffs [14] of a three-tiered grading system (System 1), which has been used in most studies to evaluate inter-observer reproducibility of TIL scoring among pathologists [18,19,30] separating the tumors into three-tiered levels of TILs, can be safe for scoring TILs in breast cancer but needs to be assessed more in CRC. To the best of our knowledge, the only studies that have evaluated the methodology of ITILsWG in CRC are those of Iseki et al. [17] and Fuchs et al. [18], and according to these studies, this methodology was a strong predictor of survival in patients with CRC.

### 4.2. TILs in the Tumor Front

According to the results of this study, the correlation among pathologists for estimating the tumor-front TILs density is moderately positive, which in more detail, the correlations between OBS2 and OBS3 were higher than the correlation with OBS1.

According to Koo and Li [27] criteria, ICC for evaluating tumor-front TILs can be interpreted as poor to moderate agreement between pathologists. The overall kappa value for the different categories of TILs density in the tumor front corresponds to a poor agreement among pathologists [28]. This study shows that, in comparison with the three-tiered level of TILs density in the tumor front, the concordance for each pathologist was improved when using a two-tiered level of TILs in the tumor front (System 4), indicating that estimating tumor-front TILs in tumors with low vs. high levels of TILs density is more reliable in clinical practice. In comparison with three-tiered categories, according to cutoff System 4, which is a two-grade categorization, between OBS1 and OBS3 who are experienced in scoring TILs, a higher agreement is observed. The kappa statistic and concordance rate showed that the ITILsWG proposed methodology [14] is slightly safe for identifying TILs in the tumor front. However, the positive consensus rate in System 1 was slightly lower than that in the other cutoff systems. System 4, which is a three-tiered grading system, is safer to use in daily practice, indicating an appropriate cutoff system for categorizing TILs density in CRC.

The percentage of cases in which two pathologists agreed to estimate the TILs density in the tumor front. Kappa statistics for all cutoff systems indicate between OBS2 and OBS3 is a fair or moderate agreement. In general, there were no significant differences between OBS2 and OBS3 in all grading systems in scoring TILs in the tumor front.

The frequencies of cases in each category of TILs in the tumor front recorded by pathologists showed that OBS1 estimated the tumor-front TILs with a higher threshold. The heat map indicates the challenging slides for evaluation that led pathologists to underestimate or overestimate TILs at the tumor front. It is more challenging for pathologists to estimate TIL density in cases with a moderate mean value of TILs in the tumor front. Given that the standardized methodology of ITILsWG has been proposed for scoring intratumoral TILs in breast cancer, few studies show using it for scoring intratumoral TILs [18] and tumor-front TILs [17] in CRC, but this study showed equal agreement can be obtained by using the methodology for scoring intratumoral and tumor-front TILs in CRC. Furthermore, Iseki et al. [17] showed that evaluation of TILs by the observer was correlated with the software assessment of TILs. Computational assessment of TILs in the tumor front can validate the usefulness of the recommended methodology and suggest cutoff values. In addition, it seems OBS1 estimated the tumor-front TILs with a higher threshold, and patterns of tumor-front TILs estimated by OBS2 and OBS3 showed that they gave lower scores. This evidence showed the source of variability in TIL scoring is the different thresholds among observers [19].

### 4.3. Tumor–Stroma Ratio

According to Spearman coefficient analysis, the correlation between OBS1 and OBS2 exhibited a higher value compared to the correlations observed between OBS1 and OBS3, as well as between OBS2 and OBS3. The ICC value was indicative of poor agreement between pathologists in estimating TSR. The ICC value indicates that more than 50% of the variance in the results can be attributed to the variance caused by non-stromal attributes; the remaining variance is due to stromal proportion differences. The overall kappa value showed moderate agreement between all pathologists in estimating TSR in CRC tissue sections [31]. In comparison with the study by Souza da Silva et al. [23], the kappa value obtained in our study showed less agreement. There are various reasons for disagreements among pathologists. Therefore, this evidence shows that to reduce the disagreement between pathologists in estimating TSR, more research is needed.

However, a good concordance rate between the pathologists was found, and a positive consensus was obtained, in 87.2% of the cases. These results show that even if the estimation of the TSR is visual and subjective, pathologists agree on approximating the percentage of the stromal area compared to the tumor area in CRC tissue. Pairwise agreement evaluation indicated that the agreement between pathologists for estimating the TSR ranged from fair to good; therefore, there was no difference between pathologists in approximating the TSR. The results vouch for pairwise agreement among pathologists in the scoring of TSR on CRC slides. Additionally, the variability in approximating the stroma-rich cases was higher than that in stroma-poor cases.

### 4.4. Correlation between Scored Biomarkers and Patient Data

To calculate the correlation between the estimated biomarker and patient data, we calculated the median value of TILs and the TSR scored by pathologists. According to the results, there was a poor-to-moderate correlation between tumor grade and the median value of TSR, but there was no evidence of correlation with the sex and age of patients. Other studies also show TSR is associated with CRC stage but it is not associated with adverse outcomes [31].

### 4.5. Limitations

This study had some limitations. First, this was a retrospective study with a small sample size. Given that Mashhad is a medical referral city in the northeast of the country, patients from different cities and places had come there to undergo surgery; in addition, fragmented health and demographic systems led to a lack of access to survival status, TNM classification, and other clinical data for some patients. Therefore, prospective studies with larger sample sizes and diverse people from different regions are required. Second, the sample size was not sufficient to clarify whether using the recommended methodology was associated with better patient clinical outcomes. Further studies are needed to validate cutoffs of TILs scored in intratumoral and invasive front according to the ITILsWG methodology to evaluate its association with patient survival and compare it with TNM classification. Third, the lack of access to IHC-stained sections made it impossible to compare TIL scores and immunoscores on HE and IHC, respectively. Finally, the observers in the study were senior pathologists who had different levels of experience in clinical and surgical pathology, and although an observer who had more experience in TIL detection answered different questions regarding the aim of the study, TILs, and TSR estimating, and recording the scores, the subjectivity of the measuring by humans remained. On the other hand, scores recorded by the less-experienced observer were correlated with the more-experienced observer’s scores. Therefore, it seems that a computerized diagnostic assistant on the HE slides can aid pathologists to overcome the subjectivity of biomarker quantification in clinical settings.

## 5. Conclusions

In conclusion, the present study confirmed that the evaluation of TILs in CRC using HE slides according to the methodology recommended by the ITILsWG provides poor-to-moderate agreement among study pathologists. Despite the discordance among observers in assessing TILs, the two-tiered cutoff system is safer to apply in routine practice because of less disagreement among pathologists. Furthermore, although the assessment of TSR using the proposed methodology is appropriate, a moderate inter-observer agreement was observed among pathologists with different experiences in pathology professions. In the future, a study that automates biomarkers’ estimation using artificial intelligence would aid pathologists in detecting biomarkers on HE-stained sections to smooth the disagreements between pathologists in the routine clinical setting. Applying multi-phasing training-observing to a large-scale study may be beneficial in improving the agreement between pathologists to create ground truth and help to reduce any possibility of subjectivity bias. However, larger studies with diverse populations are required before using TILs and TSR to assess CRC patients in practice settings. 

## Data Availability

All data are stored and available upon request to first author. azar.kazemi@tum.de.

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
