# Peer review of "Assessment of the Tumor–Stroma Ratio and Tumor-Infiltrating Lymphocytes in Colorectal Cancer: Inter-Observer Agreement Evaluation"

_diagnostics, 2023, doi:10.3390/diagnostics13142339_

Round 1

Reviewer 1 Report

Study design is appropriate.

This study must be conducted with larger population involving diverse people from different regions in the future. 

Is there any comparison of this system with other Cancer classification system like TNM?

Author Response

Dear Reviewer:

Thank you for taking the time to review our manuscript. Your insightful comments and suggestions were invaluable. Your feedback helped us strengthen the manuscript.

Point 1: This study must be conducted with larger population involving diverse people from different regions in the future.

Response 1: Thank you for this comment. As you mentioned, the results from this small-size study need further validation with a large-scale study. We add your comment in the limitation and conclusion sections to encourage researchers from all over the country and regions to conduct a larger study.

Point 2: Is there any comparison of this system with other Cancer classification systems like TNM?

Response 2: Unfortunately, since health systems in our country are not integrated and some patient's data have been stored only in paper records in the hospital, and this study was the first phase of a bigger study (computational pathology study), we utilized previously gathered confirmed CRC data from a previous study [1] and outcome and TNM classification were not available. These are our limitations, and thank you for this comment, we add it to the limitation section to suggest continuing this study to investigate any association between biomarkers and patient outcome and also compare it with TNM classification. Please see green highlighted parts in the limitation and conclusion sections.

[1]. Bani N, Moetamani-Ahmadi M, Alidoust M, ShahidSales S, Khazaei M, Esmaily H, Joudi-Mashhad M, Ferns GA, Gharib M, Avan A. Association between the 308 G> A variant of the TNF-α gene and risk of colorectal cancer. Meta Gene. 2021 Jun 1;28:100878.

Sincerely,

First author

Reviewer 2 Report

thank you for allowing me to review this original article. 

between 2010 and 2017, the authors only included 86 patients with stage II and III colorectal cancer. how many patients were operated on during this period and why only 86 patients were included. were there any differences between included and excluded patients? 

Of the 3 pathologists selected, their experience ranged from 8 to 17 years, and two out of three had diagnostic expertise. isn't there a bias? 

could the age of the patients, between 21 and 87, have an impact on the histological result? 

Author Response

Response to Reviewer 2 Comments

Dear Reviewer:

Thank you for taking the time to review our manuscript. Your insightful comments and suggestions were invaluable and helped us to improve the quality of our manuscript and ensure a comprehensive and transparent presentation of our findings.

Point 1: between 2010 and 2017, the authors only included 86 patients with stage II and III colorectal cancer. how many patients were operated on during this period and why only 86 patients were included. were there any differences between included and excluded patients? 

Response 1. 101 cases with histological features of colorectal adenocarcinoma and available HE-stained sections were collected. These cases were not total operations during 2010-2017, we utilized previously gathered confirmed CRC data from a previous study (reference 24), and then slides were gathered from the pathology archives. Since the availability of slides was important to us, we checked the archive and did not gather the total number of CRC cases slides. Unfortunately, we can not exactly mention the total number of patients operated between 2010-2017.  

HE slides were reviewed via light microscopy by a pathologist to select one representative HE slide (4 µm thick) for each patient. Patients were excluded if they were treated before surgery, if no tumor mass or invasion was found in the tissue section, or if the tumor was entirely necrotic. Finally, the study cohort included eighty-six cases in whose HE slides were assessable.

So availability of slides and other criteria mentioned above were considered to include or exclude cases. We edited the method section. Finally, the study cohort included eighty-six cases that their HE slides were assessable. Please check highlighted paragraph in 2.1. Patients section.

  1. Bani N, Moetamani-Ahmadi M, Alidoust M, ShahidSales S, Khazaei M, Esmaily H, Joudi-Mashhad M, Ferns GA, Gharib M, Avan A. Association between the 308 G> A variant of the TNF-α gene and risk of colorectal cancer. Meta Gene. 2021 Jun 1;28:100878.

Point 2: Of the 3 pathologists selected, their experience ranged from 8 to 17 years, and two out of three had diagnostic expertise. isn't there a bias? 

Response 2. Thank you for this comment. To address the concern raised about bias in the selection of pathologists and their varying levels of experience, we have considered this concern and would like to provide a thorough explanation to address it. The changes have provided in Material and Method section of the  revised manuscript. It is important to note that all three pathologists involved in our study are senior profesionals with significant experience in the filed. Their experience ranged 8 to 17 years and two of them were more familiar with TILs detection. We believed that having a team of senior pathologists mitigates possibility of bias compared to having one or two junior pathologists. The inclusion of highly experienced pathologist ensure a more reliable and accurate assessment of the biomarkers. Moreover, within our team, two pathologists (observer 1: OBS1 and observer 2: OBS2) were specifically experienced in TIL scoring, while the third pathologist (observer 3: OBS3) had less experience in counting TILs. Surprisingly, we observed a correlation between the scores recorded by OBS3, who had less diagnostic and TIL scoring experience, and the scores recorded by OBS2, who had higher diagnostic experience and previous exposure to TIL quantification. This further supports the reliability of our scoring.

Furthermore, it is important to acknowledge that TILs and tumor-stroma ratio (TSR) estimations are relatively new markers that have been introduced recently worldwide. To address the potential subjectivity bias, we implemented a question-and-answer phase during the initial stage of our study. This phase allowed us to clarify any concerns and ensure consistency in scoring TILs and estimating TSR among the pathologists involved. Similar practices have been observed in previous studies, such as Souza da Silva et al. (reference 23) , where interobserver agreement was assessed among pathologists with varying years of experience (1 to 20 years), and a brief session was conducted to introduce the scoring proposal. The concordance observed among these pathologists was found to be substantial.

Please see highlighted parts in Material and Methods (2.1 Patients) and discussion section.

  1. Souza da Silva RM, Queiroga EM, Paz AR, Neves FF, Cunha KS, Dias EP. Standardized assessment of the tumor-stroma ratio in colorectal cancer: interobserver validation and reproducibility of a potential prognostic factor. Clinical pathology. 2021 Feb;14:2632010X21989686.

Point 3: could the age of the patients, between 21 and 87, have an impact on the histological result?

Response 3. Since we had little available demographic and clinical data, we investigated the correlation between scores and patient details. So according to our results, there is no association between age and TILs or TSR. But grade was associated with the median of TSR. We add this information to the manuscript results. Please check the yellow highlighted parts in Abstract, and sections 2.4. Statistical analysis, 3.4. Correlation between scored biomarkers and patient data, and section 4.4. Correlation between scored biomarkers and patient data.

Sincerely,

First author
